# Exploring the Design of Patient-Generated Data Visualizations

Fateme Rajabiyazdi*
McGill University
University of Calgary

Charles Perin†
University of Victoria

Lora Oehlberg‡
University of Calgary

Sheelagh Carpendale§
Simon Fraser University
University of Calgary

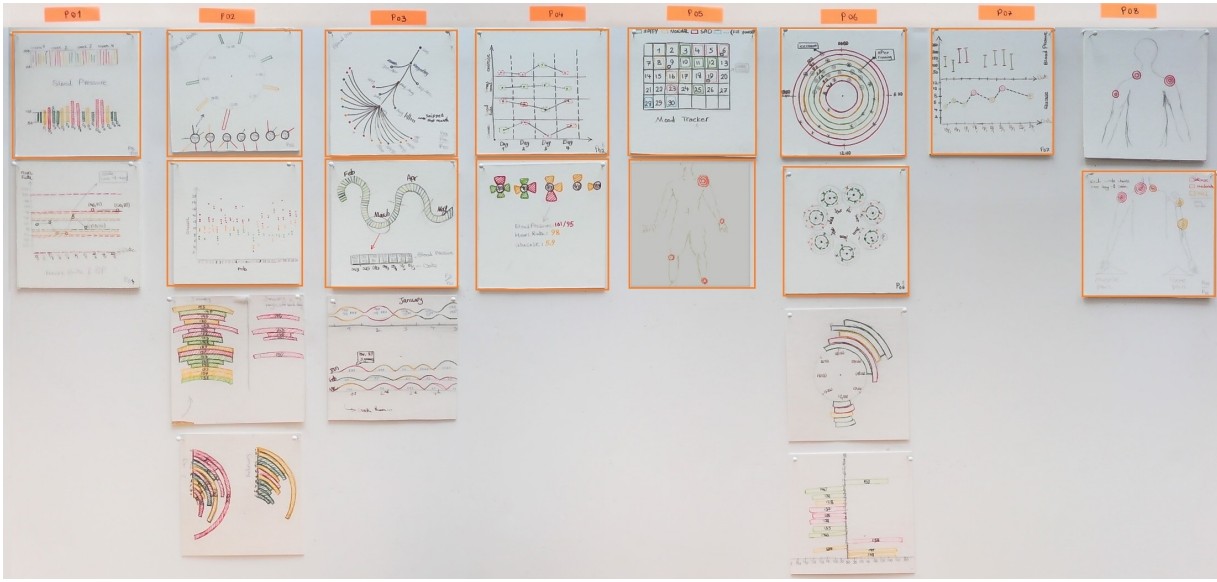

Figure 1: Design space of patient-generated data visualizations: each column corresponds to one patient and the visualizations in the column are design variations for that patient. Design variations represent the same data in different ways, thus emphasize different aspects of the data. The designs highlighted with an orange border are discussed in details in the paper.

## ABSTRACT

We were approached by a group of healthcare providers who are involved in the care of chronic patients looking for potential technologies to facilitate the process of reviewing patient-generated data during clinical visits. Aiming at understanding the healthcare providers' attitudes towards reviewing patient-generated data, we (1) conducted a focus group with a mixed group of healthcare providers. Next, to gain the patients' perspectives, we (2) interviewed eight chronic patients, collected a sample of their data and designed a series of visualizations representing patient data we collected. Last, we (3) sought feedback on the visualization designs from healthcare providers who requested this exploration. We found four factors shaping patient-generated data: *data & context*, *patient's motivation*, *patient's time commitment*, and *patient's support circle*. Informed by the results of our studies, we discussed the importance of designing patient-generated visualizations for individuals by considering both patient and healthcare provider rather than designing with the purpose of generalization and provided guidelines for designing future patient-generated data visualizations.

**Index Terms:** Human-centered computing—Visualization—Visualization application domains—Information visualization

---

*e-mail: fatemeh.rajabiyazdi@mail.mcgill.ca

†e-mail: cperin@uvic.ca

‡e-mail: lora.oehlberg@ucalgary.ca

§e-mail: sheelagh@sfu.ca

Graphics Interface Conference 2020
28-29 May

## 1 INTRODUCTION

Collecting patient-generated data is becoming increasingly common in chronic disease management [20]. Patients use technological tracking tools to collect health and lifestyle data in disparate places [3]. Both healthcare providers and patients agree that this data could be used to make smarter decisions to improve patients' quality of life and to aid providers in making decisions about patient ongoing care [36, 44, 57]. There are already technological tools for tracking and visualizing health data such as sleep (e.g., [10]), physical activity (e.g., [16]), variations in weight (e.g., [37]), and blood sugar level (e.g., [6]). However, most of these tracking tools are not designed to fully meet patients and healthcare providers' expectations [11] and do not support reviewing patient-generated data with healthcare providers during clinical visits. One way to support patients in presenting their data with healthcare providers is to visualize the patient-generated data collections effectively. Yet, we lack an understanding of *what type of visualization designs can support chronic patients to present and review their health data with healthcare providers during clinical visits.*

To answer this question, we *explored* patients' and healthcare providers' perspectives on presenting and reviewing patient data. To extract healthcare provider requirements when reviewing patient-generated data during a clinical visit, we conducted a focus group with a mixed group of healthcare providers. To uncover patient stories and their approaches to tracking and presenting their health data, we interviewed eight patients with chronic conditions who actively track their health data. Our findings revealed four factors shaping patient-generated data: data items & data context collected by patients, time commitment invested by patients to track data, patients' motivation for collecting data, and patients' support circle.

Considering these four factors, we *designed* various visualizations representing patient-generated data collections we gathered from our patients. Instead of pursuing a single generalized visualization design, we designed individually tailored visualizations for each patient. Based on our preliminary visualization designs, we proposed a design space of patient-generated data visualizations. Next, using these individually tailored visualization designs as elicitation artifacts, we interviewed the healthcare providers who had initiated the request for this project to *reflect* on the designs. Healthcare providers pointed to four use cases that they envision these visualizations could support their practice.

As a whole, the results of all our studies led to one message: the importance of designing patient-generated data visualizations by considering each patient and healthcare provider rather than designing for generalization. However, it may seem impossible to either design a unique set of visualizations for each patient or expect patients to design their own visualizations. We, as healthcare technology designers, need to provide patients and providers with a set of visualization designs as starting points. This approach would let each patient and provider choose the visualization designs that work the best for them with the capacity to customize the designs based on their lifestyle, conditions, collected data, and patient-provider relationships. Our contributions in this paper are as follow:

1. We identified four factors shaping patient-generated data.

2. We presented a design space of visualizations representing patient-generated data collections.

3. We provided considerations for designing future patient-generated data visualizations.

## 2 PATIENT-GENERATED DATA

In this section, first we discuss patients' perspectives and goals for collecting their health data. In the second part, we provide an overview of healthcare providers' perspectives on the benefits and the challenges of using patient-generated data in their practice. In the last part, we discuss how technological and visualization tools can support patients and healthcare providers with presenting and reviewing patient-generated data collections.

### 2.1 Patients' Perspectives

The number of patients with chronic conditions is increasing every day in the world. The nature of chronic conditions requires close monitoring and self-managing care for these patients [21]. A survey study in 2013 showed at least seven out of ten adults in the U.S. track a health indicator for themselves or for someone whom they take care [20]. An increase in the availability of wearable sensors, mobile health apps, and novel portable technologies provided patients with an extra boost to track more personal health data [2]. People track their health data in various forms, including memorization, original artifacts, personal paper records, personal electronic records, and electronic patient portals [3, 32].

Patients track different types and amount of data depending on their personal health goals. These goals can range from preventing more complications, having more control over their health [14, 21, 35], setting personal health goals [49, 56], improving their conditions [26, 27], and sharing these self-collected data with their healthcare providers [49, 56].

Many patients share their self-collected health data with their healthcare providers during clinical visits seeking tailored medical advice [14,21,57]. Studies have shown that sharing patient-generated data with healthcare providers can improve patient-provider communication [36,44]. Sharing health data also empowers patients in taking control of the conversation during a clinical visit and helps healthcare providers build a relationship with patients [34].

### 2.2 Healthcare Providers' Perspectives

Some healthcare providers see value in patients collecting their health data and presenting them during clinical visits. They think that by reviewing patient-generated data, they will gain more insights into patient goals and will be able to provide more tailored care to patients [25]. Providers think, in some cases, patient-generated data might be more reliable than clinic measurements because the data is collected at more frequent intervals, and there is less recall bias [25, 33]. Providers mentioned that often, a hospital's electronic medical record system have misinformation or inaccuracies. In addition, patient data measured in the clinic (such as blood pressure) may be affected by the white coat effect and stress of the clinical environment [41, 51]. In these situations, patient-generated data can be used to reconcile these inaccuracies [25] as patient-generated data may contain less false-positive data than patient health data collected in the clinic.

We should note that although healthcare providers may find patient-generated data complementary to clinical measurements and history taking if tracked in a meaningful way, they do not consider this data as a replacement to clinically measured data [33]. Patients may not be willing to record their data when they have abnormal readings due to fear of consequences [4] or may be worried that their data will be part of their permanent clinical record [17]. In addition, providers sometimes express frustrations when patients do not track enough data, track excessive data, or track non-meaningful data [4, 39]. Patients also use different mediums and organization formats that work best for them to collect and present their health data. As a result, the patient-generated data collections become heavily personal and complex, making it challenging for healthcare providers to understand and analyze [3].

It is difficult to find the time to examine unrequested data during a short clinical visit [40]. Most clinical visits are currently short [18]. The common clinical visits with family physicians usually last about 10 to 20 minutes [23], leaving a short amount of time for reviewing patient-generated data [13]. The providers may not find as much value reviewing patient-generated data during a clinical visit [42]. Storing this data safely and securely can be challenging for providers and can add to their workload [50]. Thus, there is still not a fully clear understanding of what type of patient-generated data is most useful to review and discuss during clinical visits.

### 2.3 Summary

One way to facilitate reviewing patient-generated data would be to have standardized data collection and presentation processes [13]. However, a standardized process is probably not a panacea, as every patient and healthcare provider may have individualized preferences and needs [9, 39].

There is evidence that technology can support providers and patients in improving the quality of communicating patient data [47,55]. Previous work raised questions about how technology should be designed that could assist both patients and healthcare providers in smoothly reviewing patient-generated data during clinical visits [57]. One way could be visualizing these patient-generated data. Visualizing this data can benefit both patients and providers, if carefully designed so that it seamlessly integrates both perspectives into patient care planning [39, 40].

However, the question of how and what type of visualizations to design for chronic patients to present and review their health data with healthcare providers during clinical visits remains not fully answered. Designing a general solution that works for all patients and providers is not easily achievable. Thus, first we need to move towards designing tailored visualizations, making an individualized visualization experience for each patient and provider.

## 3 METHODOLOGY

We were approached by a group of healthcare providers from a local hospital who are involved in the care of chronic patients to explore if, and how, to design technology that can enhance the presenting and reviewing patient-generated data during a clinical visit. To answer this question, we took an iterative design approach with involvement of patients and healthcare providers (Fig. 2). The study was approved by the University of Calgary Institutional Review Board.

First, (1) we conducted a focus group with the healthcare provider that voiced concerns for reviewing patient-generated data. Then, to complete the healthcare providers' perspectives reviewing patient-generated data, (2), we interviewed eight patients who actively collect health data and collected a sample of their data. We asked our patient participants about their experience collecting, analyzing, and sharing their data with healthcare providers. Next, we leveraged this understanding to propose potential visualization designs representing patient-generated data that we collected. Our goal was to design visualizations to improve the process of reviewing these data during clinical visits. Last, (3) we interviewed healthcare providers seeking their reflection on our proposed visualization designs and asked how they envision using these visualizations in their practice.

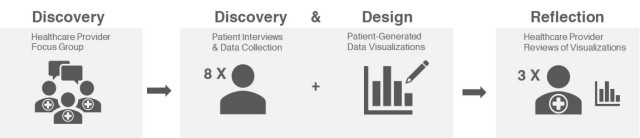

Figure 2: Methodology: Discovery, Design, and Reflection

### 3.1 Healthcare Provider Focus Group

To clarify, confirm, and gain a deeper understanding of the healthcare providers' perspectives about the patient-generated data collection review process, we conducted a formal focus group with a mixed group of healthcare providers. Our focus group included a subgroup of providers who initially approached us including a clinical endocrinologist (with 21 years of experience), one internal medicine specialist physician (with 29 years of experience), and one health counsellor and researcher (with 9 years of experience) supporting patients who monitor their data. Three other healthcare researchers were present during the focus group listening to the discussions.

In our focus group, we asked healthcare providers about their experiences reviewing patient-generated data, analyzing and understanding patient data, and giving advice to patients based on their data. One interviewer primarily posed the questions during the discussion and two other researchers from our interview team took field notes. The focus group lasted around 60 minutes. We video-recorded and transcribed the focus group discussions. Later used open-coding approach and grounded theory [46] to analyse the data in MAXQDA2012. All members of the research team (authors) participated in multiple rounds of coding and identifying themes.

### 3.2 Patient Interviews and Data Visualization Designs

To understand patients' perspectives on tracking and presenting their self-generated health data, we interviewed eight patients who suffer from one or multiple chronic conditions. We used several methods of recruitment for this study: emails from a local Patient Care Networks directors, Patient Care Networks newsletter ads, targeted recruitment through the healthcare provider who participated in the focus group and snowball sampling.

We conducted an hour long semi-structured interview with each patient. We formed our patient interview questions based on the results of our discussions during the focus group with healthcare providers. We asked participants to bring a sample of their data to

the interview session and walk us through their data sample in detail. We video-recorded and transcribed all the interviews. We used open-coding approach and grounded theory method [46], analyzing each interview in a separate process in MAXQDA2012. Our goal was to reach a deeper understanding of each patient's story. We state proof of existence for each interview and do not try to generalize our findings across patients. Next, based on the requirements that were identified after analyzing each patient interview, we sketched various visualization alternatives representing that individual patient-generated data collection. One member of the team designed all the visualizations and as a group, we discussed the visualizations and how they meet the patients' needs. Then, we selected one or several alternative designs that best matched the patient's story and needs.

### 3.3 Healthcare Provider Interviews

To complete our design cycle, we took our visualization designs back to three healthcare providers, who were among the group that initiated this project, seeking their feedback. We interviewed two healthcare providers who participated in our focus group, an internal medicine physician with 29 years of experience (C1), a clinical endocrinologist with 21 years of experience (C2), as well as a complex chronic specialist physician with 22 years of experience (C3). Each session lasted between 40-60 minutes and was video recorded and later transcribed. In the session, we first gave providers a description of the patients' conditions, their personal stories, and their data collection processes. Then, we shared the visualization designs with the providers and observed their reactions walking through and talking out loud about the designs.

## 4 PROVIDER FOCUS GROUP FINDINGS

From the results of our focus group analysis, we extracted four requirements mentioned by healthcare provider participants to support reviewing patient-generated data during clinical visits.

**R1-Integrating data context:** Healthcare providers think patient sometimes collect too many data items, but data without context is not helpful for medical purposes, "*you get the data in a 7 by 6 table with numbers and they are all over the place. Without food information, stress information, activity information it does look like a bunch of noise. You don't see a pattern without being able to query on those other dimensions. Like your sugar is high, are you stressed?*" (C1). To overcome this challenge, providers need **tools that are able to integrate context with data**.

**R2-Summarizing for quick presentation of data:** Patients sometimes come to clinical visits with a large collection of data and expect their healthcare providers to help them make sense of their data "*they clearly put in a lot of work, but you don't have time and you have nowhere to begin*" (C1). Healthcare providers want **tools with abilities to summarize and filter patient data** to see trends, patterns, and anomalies.

**R3-Sharing goals and motivations:** Our healthcare providers told us patients usually have different goals than providers which may cause conflicts. Patients often like to discuss details of their data, but providers are more interested in an overview of the whole data, so they wanted "*a platform that forces people to be explicit between stakeholders*" (C2). With this in mind, providers wanted to have **tools with ability to overview and focus on parts of the data** to explore the patient data in both focused and detailed views accommodating their goals and patients' goals.

**R4-Supporting conversations:** Both patients and healthcare providers need support to discuss their concerns "*[patient says] I have questions about [this] and the doctor says ok, great, that is what is going on there. But I am more concerned about this*" (C1). Healthcare providers told us they need support opening up communications with patients which may have not happened otherwise; **tools that can represent patient data in different views letting patients and providers discuss various aspects of patient data**.

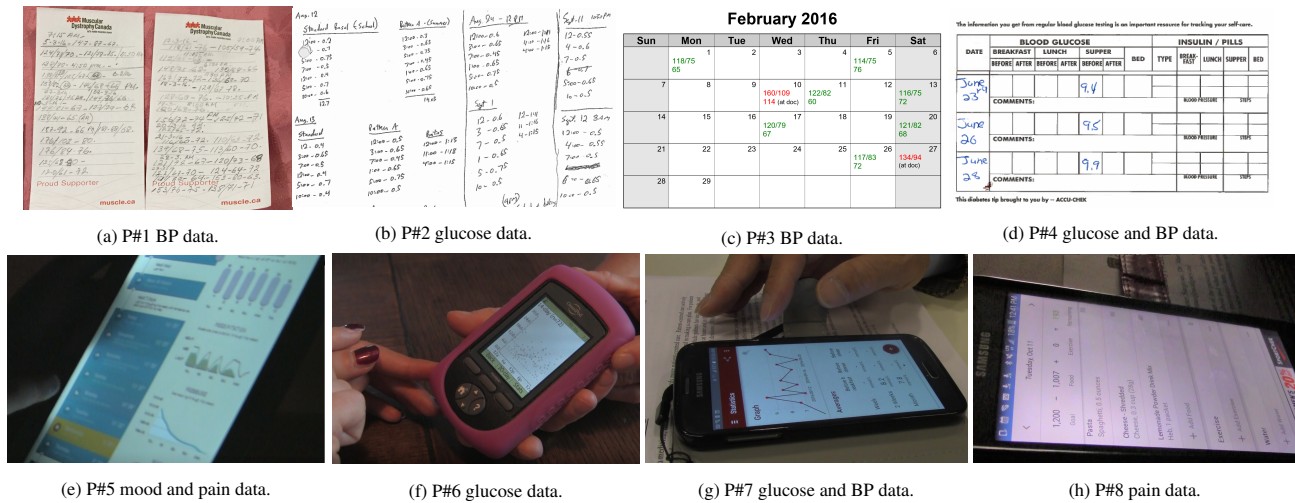

(a) P#1 BP data.  (b) P#2 glucose data.  (c) P#3 BP data.  (d) P#4 glucose and BP data.

(e) P#5 mood and pain data.  (f) P#6 glucose data.  (g) P#7 glucose and BP data.  (h) P#8 pain data.

Figure 3: Sample recordings of patient-generated data collections for patient #1 to patient #8.

## 5 FROM HEALTHCARE PROVIDER FOCUS GROUP TO PATIENT INTERVIEWS

The findings from the focus group helped us form our patient interview questions. Our healthcare providers found patient-generated data useful when patients collect meaningful data with context. Thus, in our patient interviews, we asked patients to talk about the *data* items and the *context* data they collect. Our providers expressed their concerns about patients committing an excessive amount of time on data collection resulting in large datasets. Thus, to get patient perspectives in this manner, we asked patients to tell us about their *time commitment* to data collection. Our healthcare providers talked about the impact of patient goals and motivation on their data collection and data sharing. Thus, in our patient interviews, we asked patients to tell us about their goals and *motivation* for collecting data and if they were advised to track data by their providers. Our healthcare providers saw value in having a patient's presence during clinical conversations. Thus, we asked the patients whether they shared their data with their healthcare providers or their caregivers at home and their experience with receiving *support*.

To design the patient-generated visualization designs, we considered the four requirements (R1-R4) identified from our focus group and followed the design guidelines established in the literature. To accommodate data context integration, R1, in the visualization, we used "Tooltip" which is an identifying tool presenting the attribute data attached to an object. To incorporate R2, we followed the basic information seeking principles [1]. To fulfill R3, we incorporated "overview and details-on-demand" interactions [43] in our designs. To support patients and providers view patient data from different perspectives, R4, we designed multiple visualization designs for each patient-generated data collection.

## 6 PATIENT INTERVIEW RESULTS AND PATIENT DATA VISUALIZATION DESIGNS

We interviewed eight adult patients managing one or multiple chronic conditions including hypertension, Type 1 diabetes, Type 2 diabetes, chronic pain, depression, and arthritis.

In each part of this section, we first present the patient profiles; their data and context, their time commitment, their motivation, and their support circle. We allocated pseudonyms to confer the anonymity of our patients. For each patient, we ideated and designed multiple visualizations. These visualization designs are simple visualizations that are carefully designed to capture providers' requirements and each individual patient needs and may not be novel

designs by themselves. We explain the detail of each visualization we sketched to display patient data and how we took providers' and patients' requirements into considerations when exploring visualization design opportunities to represent their patient-generated data collections. We did not restrict ourself to designing a certain number of visualization representations; we sketched as many design possibilities as we could think of to present the data for the patient.

In total, we generated 20 preliminary visualization designs for eight patients. We laid out these designs on a design space board (Fig. 1). In this design space, each column corresponds to one patient and the visualizations in the column are design variations for that patient. We acknowledge that these designs are not the only possible visualizations and other designers/researchers may come up with variations to these designs. Here, we present our designs and we hope this will be a starting point for other researchers and designers to contribute more patient stories to the literature and to move towards thinking about designing more for individuals.

Next, as a group, we discussed all of the visualization designs and selected the design(s) that best represent each individual patient based on the patient needs that were identified after analyzing the interviews. In the Fig. 1, the selected designs are highlighted with an orange border.

### 6.1 Patient #1: Maria Freeman

Maria is 67 years old and she previously had hernia and hysterectomy operations. One day she experienced high blood pressure and visited the hospital emergency room. After that hospital visit, Maria constantly experienced high blood pressure. That year, she was diagnosed with hypertension.

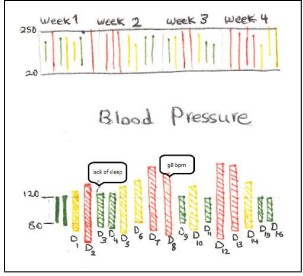

Figure 4: P#1 preliminary visualization sketch.

**Data & context:** Maria was advised to track her blood pressure and heart rate on a regular basis using a cuff machine. She uses a notebook to record her readings (Fig. 3 - a). We designed a visualization representing both Maria's blood pressure and heart rate readings (Fig. 4). We display blood pressure readings in the form of bars and show the patient's heart rate on demand. Each bar represents one blood pressure reading, we associate the bottom border of the bar to diastolic and the top border of the bar to systolic. The two horizontal lines in the background show the normal blood pressure reading range (120 over 80). In addition, we added colour to each bar showing a normal (green), an abnormal (yellow), or a dangerous (red) blood pressure reading.

**Time commitment:** Maria tracks her blood pressure and heart rate three to four times per day. Thus in our design each bar in the visualization shows one reading with the time of the recording.

**Motivation:** Maria's ultimate goal for tracking her data is *"to feel better ... make my blood pressure go down"* (P01). After her diagnosis, she changed her life style to reach her goals. She is drinking more fluids and reduced the amount of salt in her diet. She is hopeful that she can reach her goal. She also keeps a record of events or activities she thinks may be relevant to her blood pressure, so later during a medical visit, she can discuss them with her healthcare providers. Thus, in our design we have an option to add notes associated with her blood pressure records.

**Support circle:** Maria presents her notebook to her family physician saying, *"because of this [notebook], it will be easier for me to inform the doctor"* (P01). She hopes her family physician can make sense of the data and make adjustments to her treatment plans based on her data. To accommodate Maria's need for sharing her data, we designed this visualization with the capacity to show an overview of blood pressure readings over months (top row in the design) to quickly check her overall status in the past months as well as detailed numbers on demands (bottom row in the design).

## 6.2 Patient #2: Andrew Gellar

Andrew was diagnosed with type 1 diabetes about 16 years ago at the age of 52. Due to his age, he was first misdiagnosed with type 2 diabetes. After his diagnosis, his interaction with the healthcare system changed from visiting his family physicians once a year to getting an A1C test every three months. He has been in direct interaction with a nurse educator, a foot care clinic, and an endocrinologist in a diabetic neuropathology clinic.

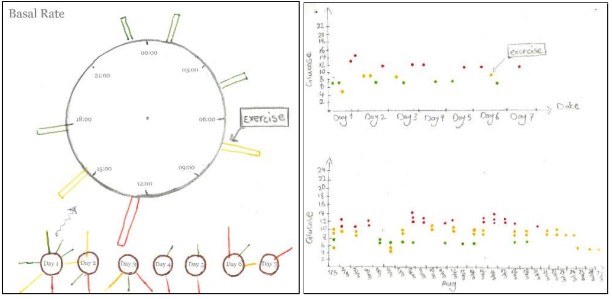

Figure 5: P#2 preliminary visualization sketches, left (A), right (B).

**Data & context:** Andrew measures his blood glucose and basal rate as advised by his nurse educator and endocrinologist (Fig. 3 - b). He uses a glucose meter to measure the concentration of glucose in his blood and an insulin pump to calculate the amount of insulin required. We represent Andrew's blood glucose data in two different visualization designs. The (Fig. 5 - A) design, shows a detailed view of one day of Andrew's glucose level. The circle shows a 24-hour clock. Each time Andrew measures his glucose, we show his reading on that time on the clock with a bar. The height of the bar represents

the glucose rate and the color of the bar represents the normality of the glucose rate; if the glucose reading is too low (red), low (yellow), normal (green), high (yellow), or too high (red). In the (Fig. 5 - B) design, the top part shows all blood glucose ratings recorded in a month with circular points. The y-axis shows the glucose rate and the x-axis shows the date. We also double coded each data point with the same colour themes as the first design.

**Time commitment:** Before each meal, Andrew measures his blood glucose using the glucose meter and enters his readings into the insulin pump. The pump automatically send Andrew's insulin intake to his nurse educator. Besides that, he keeps track of his basal rates that he measures using the glucose meter, in a notebook to later share with his nurse educator. Every time Andrew visits his healthcare providers to check his conditions, he shares the data he collected over the past few months with his healthcare providers. Thus, in our visualization designs, we included a weekly or monthly overview of his glucose rates at the bottom of both designs.

**Motivation:** Andrew lives a good life, eats healthy, gets enough sleep, and has a balanced work-life lifestyle. He recently got diabetes complications. After experiencing the complications, he is hoping to start an exercise routine. Andrew tracks his exercise on the side to understand the effect of his physical activities on his blood glucose. Thus, we added an option for the patient to add a free style note (e.g., exercise) on his data point to appear on demand when hovering over the data point in the visualizations.

**Support circle:** Andrew has a hard time analyzing and finding trends in his data to adjust his lifestyle saying, *" There's so many factors that come to play with your blood sugars and trying to get everything in the right spot"* (P02). He expects his healthcare providers to make sense of his data for him and give him direct instructions on how to better manage his conditions. Thus, we included a weekly and a monthly view of the glucose recordings on the bottom of both designs to give an overview of his data.

## 6.3 Patient #3: Jen Adams

Jen is 34 years old and was diagnosed with hypertension when she was 18 years old and was medicated for a few months. After she got off medication, she started monitoring her diet and adjusted her life style. Last year, she had a visit with her family physician to get treatment for an infection and her blood pressure reading was high at the clinic. But, when she checked her blood pressure at home, she noticed that her reading was closer to normal than in the clinic.

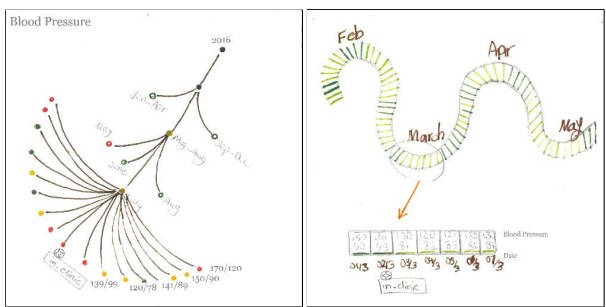

Figure 6: P#3 preliminary visualization sketches, left (A), right (B).

**Data & context:** Jen tracks her blood pressure and heart rate (Fig. 3 - c). Since she is experiencing a steady heart rate, she mainly focuses on her blood pressure data. Thus, we designed two visualization alternatives displaying Jen's blood pressure data. In the (Fig. 6 - A), we designed a tree based visualization with the ability to expand on demand. The top root represent the average blood pressure readings of the patient over one year. The next level shows the seasons, then months, and lastly the daily blood pressure reading. Jen uses three different colors to distinguish her readings into normal,

borderline, and abnormal. With colour coding her numbers, she can quickly glance over her data. We have used the same idea in our visualization design and color coded her blood pressure readings. In the (Fig. 6 - B) design, each bar shows an average of all Jen's blood pressure readings in a day, where the colours indicate the normality of the number. Dark green indicates high blood pressure readings, green indicates a normal blood pressure readings, and light green indicates low blood pressure readings. Looking at this view, she can decide if she is having more dark or light colors in a period of time. Whenever she decides to focus on a certain period of time, she can select that section and a table view appears underneath with data displayed for each day.

**Time commitment:** Jen has been measuring her blood pressure a few times per week for a year since she believes her condition is under control with steady blood pressure readings: *"Lately, it's been quite good for the last several months. So, kind of since January I check it maybe once a week now as opposed to every day"*(P03). Thus, in our designs we only display maximum one reading per day.

**Motivation:** Since her last clinical visit, Jen monitors her numbers to prevent any complications or developing hypertension for the second time. Last time she was taking medications for her hypertension, she experienced many side effects, and she fears that the healthcare providers may medicate her again: *"I've been borderline and they've talked about medicating me for it, but I would rather not be if I can avoid it. So, I am just trying to manage it other ways before getting to that point"* (P03).

**Support circle:** Jen usually does light exercises, gardening, or takes short walks to stay healthy. To stay under 1500 mg sodium per day she plans her weekly meals with her husband. She expressed her concerns to her physicians that she only has high blood pressure when she is at the clinic since visiting her providers gets her anxious and stressed. To overcome this problem, she writes notes next to her readings keeping track of any triggering factors such as 'in clinic'. She is hoping by showing the numbers she tracked at home to her healthcare providers, she can tell them, *"No, it's usually right around 120/80. It's not always this high"* (P03). Therefore, in our designs, we have an option for Jen to mark the blood pressure readings measured during her clinical visits.

### 6.4 Patient #4: Lucas Ford

Lucas is 43 years old and suffers from hypertension, type 2 diabetes, and depression. Lucas was hospitalized a few times with suicidal thoughts and high blood glucose. Tracking his blood pressure and glucose level helps him get his conditions under control; however, sometimes he experiences an emotional break down when his readings are higher than the normal range advised by his providers.

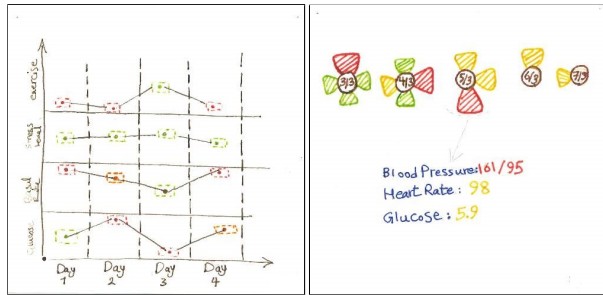

Figure 7: P#4 preliminary visualization sketches, left (A), right (B).

**Data & context:** Lucas collects his glucose, blood pressure, and heart rate data (Fig. 3 - d) in his notebooks. Lucas wants to look at his glucose, blood pressure, and heart rate data all at once. Thus, we designed two different visualizations displaying all three items he is tracking (Fig. 7) in one view. In the (Fig. 7 - A) design, each

vertical division in the chart shows one data item: blood pressure, blood glucose, and heart rate. In the (Fig. 7 - B) design, we show each day of data readings in a flower shape visualization, each petal representing one data item: blood pressure, blood glucose, and heart rate. Each data point is color coded in both visualizations based on the ranges defined for Lucas's conditions. Green indicates normal, yellow shows borderline, and abnormal readings are colored in red.

**Time commitment:** Lucas was advised by his providers to record his data five times a day. However, he is dealing with a lot of pressure due to his conditions and his personal problems, so he only manages to track his data once a day.

**Motivation:** He feels frustrated and upset with himself for not having his conditions under control. Lucas hopes to get support that motivates him to track his data, but does not want to be pushed. He wants to exercise regularly, as it can help him stabilize his blood pressure and glucose level; however, his busy schedule does not allow for exercise. Instead, he tries to go for short walks to lower his blood pressure when he experiences high blood pressure. His goal is to get off the insulin by next year.

**Support circle:** Lucas feels that he does not have enough family support and his family lacks compassion and doesn't understand the seriousness of his conditions. He has difficulty making sense of his data and expects his healthcare providers to understand his data and give him advice based on them. For instance, he was hoping to find relations between his blood pressure readings and glucose level, but could not find any correlations. Thus, to support Lucas find relations between his data, we visualized all the three data items he collects adjacent to each other in one view.

### 6.5 Patient #5: Ken Smith

Ken is 37 years old and suffers from multiple conditions. He had memory problems, paranoia, and learning difficulties since childhood (1986). He was diagnosed with behavioral disorder in 2005, mental health problems in 2009, and asperger syndrome in 2011. In addition, Ken has digestive problems and is experiencing pain in different parts of his body (e.g., neck, back, shoulder, ankle), which have not been officially diagnosed.

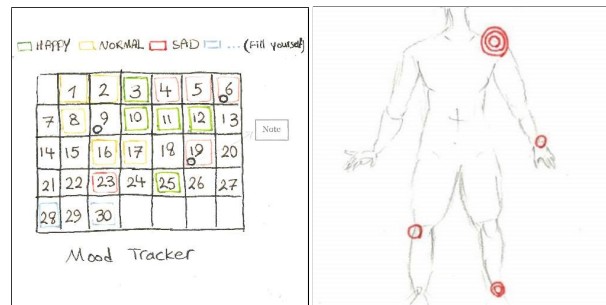

Figure 8: P#5 preliminary visualization sketches, left (A), right (B).

**Data & context:** Ken tracks his nutrition data and symptoms related to his stomach pain and bowel movements using MySymptoms app. He tracks his pain to help with diagnosing the source of his pain (Fig. 3 - e). To understand the effects of his mental state on his conditions, he also tracks his mood. Ken prefers using multiple apps on his tablet to record different health data items, therefore we also visualized his data in separate designs. He is happy with the app he uses for tracking his nutrition, so we focused our designs on the other data items (moon and pain). We sketched a visualization displaying Ken's mood data (Fig. 8 - A). Each day on the calendar shows Ken's mood of the day which is colour coded; happy (green), normal (yellow), sad (red), and self-defined (blue). We sketched a second visualization representing Ken's pain data displayed on a body mock-up drawing (Fig. 8 - B).

**Time commitment:** Ken tracks his mood once a day. We used a calendar visualization to present his data allowing for one mood entry per day ( Fig. 8 - A). On the other hand, the pain body mock-up visualization (Fig. 8 - B) lets Ken record his pain occurrences. Each ring in this visualization represents pain experienced once in the marked location of body.

**Motivation:** Ken's goals are to eat healthier, get more physically active, lose weight, and get more involved in his care. He takes note of his the relevant context that he thinks may trigger his mood. Thus, we added a free style note option for him to track the contexts associated with each day in the calendar view visualization.

**Support circle:** Ken tracks several symptoms and trigger factors that he thinks may be helpful for improving his health, but his healthcare providers do not always find his collected data useful. He is confused about which data items are useful to collect: *"I gave all my symptoms to her, all recorded on a sheet. She said, 'Oh, we're just looking at the gut issues.' I'm like, What about the rest?"* (P05).

### 6.6 Patient #6: Sarah Green

Sarah is 49 years old and was diagnosed with type 1 diabetes in 1984. In 2013, Sarah was hospitalized experiencing severe gastroparesis symptoms. Later, Sarah developed arthritis in her hand and gets cortisone shots, which increases her glucose level after each shot.

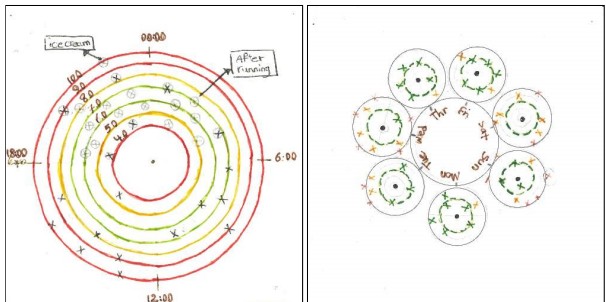

Figure 9: P#6 preliminary visualization sketches, left (A), right (B).

**Data & context:** Sarah uses an insulin pump to manage her diabetes (Fig. 3 - f). Since she has an insulin pump, the device automatically tracks her blood glucose many times in a day. Thus, to visually show all the data points measured by her insulin pump in a day, we designed a clock visualization (Fig. 9). The clock view can show all the data readings in one view with their timestamp. In the (Fig. 9 - A) visualization, the blood glucose reading is marked with an X inside each ring. The rings are colour coded to green, yellow, and red based on the ranges predefined by her healthcare providers.

**Time commitment:** The pump automatically tracks her blood glucose level in different time intervals to program her insulin. Sarah does not regularly record her food intake, but when she feels sick, she takes notes in her phone of what she ate and her activities that may have affected her glucose: *"there's really no answer, I've been dealing with this for about two or three years now "* (P06).

**Motivation:** Sarah has changed her lifestyle especially after her diagnosis with gastroparesis. She takes an active role in managing her conditions. She says *"with gastroparesis there's no medication, there's no cure ... it's a matter of just doing a lot of research and reading in different avenues* (P06). Sarah has a fear of getting sick to the extent that she needs hospitalization.

**Support circle:** Her diabetes nurse monitors Sarah's glucose level regularly. On the occasion that Sarah feels sick or in need of help, she calls her nurse and asks her nurse to log into her pump results remotely. Based on her pump results, the nurse will give her advice on how to normalize her glucose level. To discuss her readings over a week with her nurse, we displayed an overview in form of seven rings (days) (Fig. 9 - B).

### 6.7 Patient #7: Tim Muller

Tim is 56 years old and was diagnosed with type 2 diabetes about 8-10 years ago and has been also dealing with hypertension for a long time. His condition has gotten worse in the past two years. He also has a genetic disorder, Hereditary Hemorrhagic Telangiectasia that cause abnormality in blood vessel formation, but it does not affect his chronic conditions.

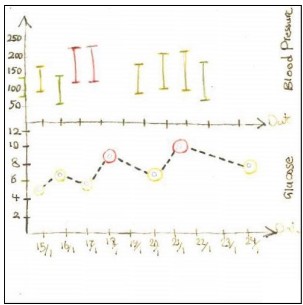

Figure 10: P#7 preliminary visualization sketch.

**Data & context:** Tim uses a glucose meter to measure his glucose reading and records his readings in an app on his phone (Fig. 3 - g). He also uses a blood pressure cuff machine to measure his blood pressure. He then manually enters his blood pressure readings into two different apps on his phone. He prefers to collect his data on his phone rather than the booklet he was given by the nurse. Since he manages two conditions, we display both blood pressure and glucose in one view (Fig. 10). Tim takes notes keeping track of special events (i.e., holidays and parties); to accommodate recording these notes, we added an option in our design to track notes.

**Time commitment:** He measures his blood glucose and blood pressure a few times a day. Thus, we show multiple data readings on the chart per day. Tim normally skips tracking his data during vacation times. However, not tracking his data during his last vacation caused an abnormality in his data: *"I was good for a while. Then took a vacation and, whoaa!"*(P07). To visually display the effect of not tracking data we show the missing dates with dashed lines.

**Motivation:** After visiting a new physician, the physician changed Tim's hypertension medication. Since the change of his medication, his blood pressure has been generally stable and he got motivated to start tracking it, *"I kicked myself, I should have tracked it longer"*(P07). He is hoping . Tim has a standing order from his diabetes nurse to get A1C test every three months. He is hoping to become more active in his care and reduce his glucose level to below 6.5: *"six months ago, it was 8.1. Now it's 7.1"*(P07). To make it easier for him to check if his numbers are normal, we colour coded (green, yellow, red) the data points.

**Support circle:** Tim's diabetes nurse and his family physician automatically receive the results of his A1C test. However, Tim does not share any of his self-collected data with his providers.

### 6.8 Patient #8: Katy Mok

Katy is 52 years old and she suffers from hypertension, asthma, arthritis, chronic pain, and depression. She was diagnosed with asthma 21 years ago which is mostly under control with medications. In 2004, she gave birth to a premature baby and had a sudden death in her family. Later that year, she was diagnosed with severe depression and was hospitalized in the psychiatric ward.

**Data & context:** As a result of Katy's depression, she gained 150 pounds. Three years ago, she joined a weight management group and was advised by her dietitian to track her food intake (Fig. 3 - h), but she does not like to share her food data with her providers. A few years ago, Katy started to experience pain in her upper body; however, her physician did not believe her pain was real

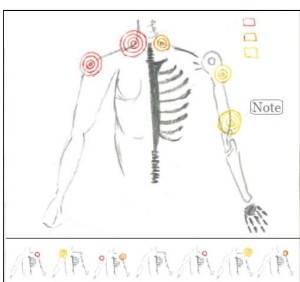

Figure 11: P#8 preliminary visualization sketch.

and was dismissive to her condition. After struggling with pain for a while, she decided to look for another pain specialist. She drew a table with upper body part names and each day she would put in a number corresponding to her pain level in addition to the type of pain (stabbing, stinging, and shooting). Thus, we designed an upper body mock-up drawing visualization to help her visually track the type and location of her pain (Fig. 11). In the visualization, the tensity of the pain is represented by the number of rings (1 to 10) and the types of pain are distinguished with colours.

**Time commitment:** Every time Katy experiences pain, she records her pain data. Thus, we also allow for as many (pain) data entry as pain occur during a day in our visualization design.

**Motivation:** Katy writes side notes to her pain data to investigate if there is any relationship between the time of the day, her activities, and her pain level. She shared her pain diary with her new pain specialist; Katy told us her specialist said: *"This is great, there is no relationship to anything which just tells me it is probably a nerve or something. This is fabulous and I want to keep this!"* (P08).

**Support circle:** Katy hopes to receive more tailored care by sharing her self-collected health data with her healthcare providers. She sees value in tracking her health data and sharing them with her healthcare providers. We display an overview to her pain data by displaying a week of her pain data in form of small body mock-ups at the bottom of our design. This view will help providers to get an overview and to find possible patterns or trigger factors.

## 7 PROVIDERS INTERVIEW RESULT

We presented the patient-generated data visualizations to three of the healthcare providers who initially requested visualizations and technological support for reviewing patient-generated data. We observed providers' reactions towards our visualization designs and asked for their feedback. The providers varied widely in why, when, and how they want to use patient-generated data visualizations in their practices. We present our results according to the two themes we identified through analyzing the interview data. The themes are 1) the visualizations' use cases in providers' practice, and 2) the platforms for implementing the visualizations.

### 7.1 Use Cases for Leveraging the Visualizations

Providers envisioned different use cases for the visualizations in their practice: 1) one provider saw value in encouraging patients to use visualizations for self-experimentation, 2) two providers wanted to use visualizations in to review patient data collaboratively during clinical visits, 3) one provider thought of using visualization to support their medical judgement when displaying all patient data in one visualization, and 4) two providers found visualizations useful when displaying patient data through different lenses.

**Encourage patient self-experimentation and goal setting:** C3, complex chronic care specialist, expected visualization views that would encourage patients to do more self-experiments. He thinks particularly for chronic symptom management where there is no complete treatment to resolve the symptoms, but rather it is a matter of trying to track and manage them, experimenting to find trigger factors can be helpful for patients. The potential of self-experimenting with data can help patients find solutions to function easier in their everyday life. In particular, C3 thinks visualization designs need to have the capacity to support patients in setting goals and tracking an intervention that patients may set in their minds to control symptoms: *"For example, taking three glasses of water per day may reduce headache"* (C3). Although this provider was interested in encouraging patients to do self-experiments and set goals, C3 wanted patients to share the data collections with him. In these circumstances, the providers can help patients understand if there is a scientific correlation between variables and help patients understand the body mechanism that might explain this correlation.

**Juxtapose data for collaborative interpretation:** C1, internal medicine specialist, was cautious about patients interpreting their data juxtaposed in a single visualization view. He was concerned that juxtaposing patient data could imply a link that may not exist and falsely medicalize the relation between the data: *" the minute you put them on a shared data exhibit, it is a correlation"* (C1). Although he was not enthusiastic about presenting data items such as blood pressure and glucose level in one view, he found coupling some data points useful. For instance, when seeing the (Fig. 1 - column p#6 - first row) visualization he was keen to view patient food intake and blood sugars displayed together to investigate their relationship. Another functionality that the providers found useful was the potential to overlay data collected across different situations or days. By overlapping data, providers may be able to find patterns in patients' data. For instance, C3 was interested in overlapping patients' glucose data over a few days to find out the effect of biking for 30 minutes on patient glucose levels , *"the nature of the adjustments is very rarely a single day"* (C3).

**Offer a holistic overview to the provider:** C2, endocrinologist clinician, showed interest in a holistic visualization view of all the data items a patient collects. She found visualization designs representing all patient data items in one view useful for planning complex chronic care. For example, for displaying blood pressure and glucose level in one view she said: *" as a care provider, I can show that 'yeah, during these times these situations are really bad for you' "* (C2). She was also keen to see patient's physical activities such as steps taken per day presented in the same view to understand the effect of exercise on the patient's other health conditions. C2 was interested in having access to the patients' notes describing the context and the situation when this data was recorded. She told us that she encourages her patients to take notes of their emotional states, their meals, or any other relevant information when recording their health data. Knowing the context associated with the data, the provider has more information to make informed medical judgments.

**Understand data better through a different lense:** Providers were able to quickly adapt to visualization designs and warmed up to the idea of alternative views of data, promising for adoption in their practice. C3 appreciated the visualizations capability to display patient data different from a standard tabular format. He thought showing patient data in different forms will give patients extra support in understanding their data and taking actions towards enhancing their health: *" We never had this kind of things [visualizations] and so, this is where the notion of 'same data, different lens' becomes useful"* (C3). The providers also recognized that some of these visualization designs can be used to represent other health measurements. One of the providers who was at first skeptical of using the blood pressure tree design (Fig. 1 - column p#3 - first row), after reviewing and discussing the design, suggested using this visualization for collectively displaying 24-hour blood pressure cuff machine data. Providers use these machines to closely monitor the patient blood pressure to help with diagnosis: *"this is an attractive idea, maybe this kind of visualization can be used for a 24 hour report [showing data] every 10 minutes"* (C1).

## 7.2 Visualization Implementation Platforms

The choice of visualization platform can make a difference in designing the right patient-generated data visualization. The providers talked to us about their preferred patient-generated data platforms, and the rationales, benefits, and trade-offs of their choices. Different technology and platforms for implementing such visualizations include data booklets, websites, phone apps, and patient portals.

**Booklets:** To smoothly integrate visualizations into providers' practices, one challenge is to design the patient-generated data visualizations compatible and aligned with the current providers' practices. Providers usually give patients tabular template booklets to record data. C3 mentioned that he preferred reading patient data in these booklet format, since it is easier and faster for him to find trends. Printed visualizations in the form of booklets can be familiar and easy to use for providers, but do not support interactivity.

**Websites:** Some providers prefer to have patient data uploaded on designated websites where they could potentially integrate patient-generated data into the patient's health records. C2 thought that, if designed well, a website would be a good platform to support both patients and providers to interact with patient-generated data and see the data in different ways. However, healthcare services usually have restrictive policies for use of websites in clinical settings.

**Phone Apps:** Patients may not feel comfortable sharing all data they collected with one healthcare provider and may only be willing to share related data with a specific provider depending on their specialty. C2 thought that using a personal phone to record data could be a solution, since patients have full authority to share data they wish. However, small display real estate could cause limitations in designing visualizations that represent all patient-generated data at once. Also, sharing a small display between patients and providers during clinical visits can be difficult.

**Patient Portals:** Providers normally have a PC in their clinical rooms for taking notes about a patient's condition and recording them in a patient's healthcare portal. C1 was keen on the idea of asking patients to link their self-collected data into their healthcare portals ahead of time. He thought that having patient-generated data collections and visualizations available on the portal could not only save time, but could also be easily accessible for discussion. However, deploying visualizations into patient portals can be a long and difficult process and requires support from healthcare services.

## 8 DISCUSSION

Effective patient-generated data communication during clinical visits can help patients feel understood and support healthcare providers get all the necessary data they need to make medical decisions [26, 29]. Our objective was to design visualizations to support patients present patient-generated data for reviewing during clinical visits. The focus of our studies was on studying patients with chronic conditions and the healthcare providers who visit chronic patients.

The results of our patient interview studies revealed the individualities and the complexities of patient-generated data collections. Each patient has a unique body, a highly individualized lifestyle, a different set of goals, and a personalized patient-provider relationship. All these factors need to be considered while caring or designing for patients [19]. How can we design only one visualization solution that can consider all these differences in patients?

Providers also differed in their principle goal of using patient-generated data. This has major implications on the design of visualization. A solution that works for one provider may not work for another. This may affect the types of visualizations we consider for them and their patients. There are many driving forces for designing effective patient-generated data visualizations. It is still unclear which direction works best for both patients and providers.

In software and technology design, research, and businesses, there is often the notion of designing with a generalization mindset, *'one-size-fits-all'*. The idea of designing one software or one visualization

tool that can address everyone's problem may be appealing and cost-efficient, but it does not always bring validation [5]. Rather, it is necessary to design for particulars, *individuals* [5].

Looking into medical literature and the approaches taken in healthcare services for patient care planning, we often see one-to-one interactions between a patient and their healthcare providers in clinical visits [23, 34]. This one-to-one interaction model has been practiced for centuries in medicine and is tailored depending on the individualities of each patient and their healthcare provider. Similarly, for designing visualizations to improve patient-provider communication, we, as visualization and technology designers, should take directions from the medical literature and their practices. We should take steps towards designing individualized tailored visualizations based on both patient and provider preferences to be able to accommodate as many patient-provider communications as possible [52]. Perhaps one solution can be to start designing and developing many patient-generated visualizations tailored based on both the healthcare provider and the patient preferences [8].

Designing visualizations to support chronic patients with their self-collected data is indeterminant, or in other word a *wicked problem* [7], meaning there are no definitive solutions or limits to this design problem. There have been attempts in the literature to design visualizations representing patient-generated data for chronic conditions including, visualizing bipolar patient lived experience [45], collaborative interactive storyboards design with chronic pediatric patients [24], and photo-based visualization to support patients with irritable bowel syndrome communicate with providers [15]. Our design study is another step towards tackling this wicked problem.

Following the criteria required to conduct a rigor design study, including multiple perspectives to shape the design problem [31], we studied the perspectives of both patients and healthcare providers. We explored the healthcare provider perspectives on reviewing patient-generated data during clinical visits and the details of eight patients' approaches to tracking and presenting their health data. Furthermore, we looked into a sample of our patient data collections to understand their methods of recording data and reasoning.

A design study is not and should not be reproducible; rather the solutions proposed are one of many possible solutions [31]. Following this criteria, we designed multiple alternative visualizations for each patient. All of our visualizations together shaped a design space of variant patient-generated data representations. We understand that depending on the patient needs, the providers' expectations, and the patient-provider relationship dynamics, a different set of visualization designs could be suitable. Our solutions are one of the many possible solutions to this wicked problem.

A rigor solution to a wicked problem needs to report the process of design and analysis in a transparent way [31]. Thus, we explained the process of design and reflection of our design in detail. We hope that the detailed design process we provided supports other researchers and designers to further tackle this wicked problem and to design patient-generated data visualizations.

Our study has limitations. Considering the challenges of recruiting healthcare providers, we could only interview 3 physicians in our study. Consequently, the healthcare providers' perspectives we provided in this paper are not generalizable. Instead, we show the disparity between healthcare providers' perspectives and draw researchers and visualization designers' attention to these differences. We interviewed 8 patients with chronic conditions; thus, their perspectives may not be representatives of all chronic patients. However, similarly to healthcare providers, patients had various perspectives that should be considered when designing for them. Lastly, we could not reach out to the same group of patients we interviewed to get their feedback on the designs. Several patients were not well enough to participate, we lost touch with some patients, and 2 patients agreed to participate. Since we only had two responses, we decided not to conduct evaluation studies with patients.

## 9 DESIGN CONSIDERATIONS

There is no single design that can fit every patient and healthcare provider; rather, visualization designs need to be tailored for each patient and their healthcare providers. To support researchers when thinking of designing patient-generated visualizations, inspired by the results of the healthcare providers' focus group, the patient interviews, and the healthcare providers' reflections on the designs, we provided the following design considerations. These design considerations are not provided to dictate future visualization designs; rather, they need to be adapted depending on each patient and their healthcare providers. Lastly, we understand that these design considerations are not exhaustive and there is room to explore other influencing factors.

**Include data context in the design:** Patients often track a large amount of data. To gain valuable insights from this data, the patients sometimes take notes of the events, circumstances, or emotions associated with the data points. On the other hand, healthcare providers in our study found this contextual information useful to make medical decisions. Previous studies also pointed to the importance of relevant dimensions and context of data for making medical conclusions [53, 54]. Thus, visualization designs need to allow for smooth inclusion of the contextual data often in forms of free-format text along with the data points.

**Consider patients' time commitments in the design:** Chronic patients often deal with many medical issues in their everyday life, leaving them with less free time to track and record their data regularly. The Apps available on the market do not usually consider patient differences in the time they invest in collecting data [48]. For instance, displaying empty entries can cause mental effects, making patients feel they are not doing enough [12]. Thus, visualization designs should allow patients to customize the amount of information and input fields being shown.

**Allow patients to freely explore their data:** Our results showed that the patient's motivation for tracking and presenting their data to providers play an important role in the design. Some patients are eager to find correlations between their data items, some patients are looking for causation of their symptoms, and some patients want to have an overview of their numbers. Previous work also stressed the need to support patients in sense-making and problem-solving with data [22, 28, 38]. Thus, these differences in patients' motivations for data collection should be considered when designing visualizations to represent patient-generated data.

**Support patients' needs to (partially) share their data:** Patients differ in the level of support they receive from their family, friends, and the healthcare team. Some patients benefit from sharing their whole data with their support circle, some patients are interested in sharing a selection of their data, and some patients are hesitant to share any data. Thus, visualization designs should support sharing overviews, selective views, and protected views. Visualization designs that support sharing views need to also include annotation capability and multiple views (e.g., patient view and healthcare provider view) [30].

**Support healthcare providers interacting with patient data:** Although healthcare providers had different perspectives on the use cases of patient-generated data visualizations in their practice, they had commonalities in regards to necessary interactive functionalities. All healthcare providers in our study talked about the difficulties of coping with messy, inconsistent, and complicated data collections. This suggests that at-a-glance data comprehension is an important visualization design goal. In addition, providers needed interaction functionalities to better understand the data including filtering the data, focusing into data details, and overlaying different parts of the data for comparison [1, 43].

## 10 CONCLUSIONS

In recent years, we have seen growing interests among patients with chronic conditions to track and analyze their data. However, sharing this data with healthcare providers can be challenging due to limited time in clinical visits and the large and complex nature of patient-generated data. We responded to a group of healthcare providers' call from a local hospital to design potential technological solutions to address the challenges of presenting, reviewing, and analyzing patient-generated data collections.

We first gained healthcare providers' perspectives via a focus group. Then, we took an in-depth look at chronically ill patients' perspectives tracking their health data. The individual differences among these patients promoted a design space approach where we used insights from these patients to design a space of possible tailored visualizations. Taking these visualizations back to the healthcare providers who made the call revealed that each provider had different ideas, purposes, and processes about how they might use these visualizations in their medical practice. As a whole, the results of our series of studies led to one message: the importance of designing patient-generated data visualizations for individuals by considering each patient and provider rather than designing with the purpose of generalization.

By exploring the possibilities of designing individual tailored visualizations representing patient-generated data, we have added one way that can support patients and healthcare providers when reviewing patient-generated data during clinical visits. We hope our proposed visualizations provide patients and healthcare providers better opportunities to present, review, and gain insights on patient-generated data. We note that we included the perspectives of a small number of patients and healthcare providers; thus, other perspectives may not be included in our results.

However, we envision this study as a stepping stone for the call to focus more on designing technologies in healthcare for individuals. We encourage the human-computer interaction, visualization, and healthcare communities to repeat these studies by including more patients and healthcare providers and explore designing tailored visualizations for each individual. Then, as a community, we can move towards accumulating these perspectives and designs to empower individuals with accessible design variations. We hope that in the long term, the results of this exploration contribute to supporting patients' and healthcare providers' in reviewing patient-generated data collections using visualizations during clinical visits.

### ACKNOWLEDGMENTS

We would like to thank all reviewers, patients, and healthcare provider participants for the expert knowledge they brought to this project. This work was supported in part by the Ward of 21st Century, AITF, NSERC, and GRAND.

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
