# OpenReview forum: "Exploring the Design of Patient-Generated Data Visualizations"
_graphicsinterface.org/Graphics_Interface/2020/Conference — GI 2020_

### Official Review · AnonReviewer3 · 2020-01-08
**Well-written qualitative paper, with only a few clarifications needed.**

**Confidence:** 4
**Rating:** 7

**Review:**

This paper describes the exploration of designing data visualizations of daily medical records by patients, and what kinds of visualizations may assist providers in best keeping track with an patient’s medical status. The authors perform three phases: An interview with providers to assess their needs, sessions with patients to gather their unique medical history and develop several visualizations for their data, and going back to providers with these visualizations to gather their ideas of how well these visualizations would assist them. The authors then suggest some design guidelines at the end for developing usable patient data visualizations.

I enjoyed the paper. It is a qualitatively-driven paper, but I believe it provides much insight into what providers would like in patient visualizes, and takes into account how patients already record their information. The writing is clear and the paper is easy to read. There are a few comments I have about the paper that I describe below.

The description of each patient drags on a little long, and much of it does not become useful after in the later sections, since particular medical history is not referenced in later sections. While identifying the uniqueness of each patient’s medical conditions and how/why they record information is important, I think this could be greatly shortened to the most pertinent points to demonstrate the differences. I would have also liked to see some of the images of the visualizations for myself.

Another concern I have is about the disparity between the emphasis on how each patient’s medical history (and in turn, visualization) is unique, and then the proposal of general design guidelines for creating patient visualizations. It seemed that the initial statement was that general guidelines were not useful because of the uniqueness at each patient. I would have liked a little more discussion on the limitations of the author’s proposed guidelines at the end and how did or did not mitigate this issue.

I think these changes/clarifications can be made easily, and therefore I would argue for the acceptance of this paper pending these changes.

---

### Official Review · AnonReviewer2 · 2020-01-08
**Exploring the Design of Patient-Generated Data Visualizations**

**Confidence:** 4
**Rating:** 6

**Review:**

The paper explores the possibilities of reviewing and visualising patient-generated data from a range of stakeholders – consisting mainly of healthcare providers and patients. The authors utilise a range of methods in order to better understand the attitude and perspective of both participants to provide relevant and appropriate design insights for developing tools to support the visualisation of data collected during a clinical visit.

First, the authors attempted to identify a gap in the literature concerning how visualisation designs can support the review and analysis of user-generated data. What is missing is a clear articulation of the research problem and question within the literature provided. It read like some form of a haphazard account of few studies that point to the relevance of tracking and visualising patient data in order to inform better health decisions, and ultimately a better lifestyle. Although sections 2 attempts to situate the research question into the context of varied perspectives, a better justification of the stake for the field would have been made clearer had it being the section doesn't read as if it’s an analysis of prior data, and not of related works.

Second, I particularly appreciate the authors' use of different methods (focus group, interviews, and observation) but fail to see an understanding of the needed sensitivity towards participants with some form of a chronic condition. It’s well known that 'chronic conditions might take a different form and thus interpreted within a particular context; this makes the contribution of the paper marginal, as one would expect a clear articulation of how the method is chosen to fit into the context of the wider literature on similar issues and ultimately the nature of the study participants. We need more detail to determine whether what the data suggest reflect the subjective perspective of the different users that participated in the study.

Thirdly, from the discussion of the findings, quotes appear unpacked. How representative is it, what’s the bigger picture, can it be generalised to other not known scenarios? The analysis of the patient's interview provided a bigger picture of the different perspectives, and which makes the different factors more relational and understandable. Overall, the analysis lacks clarity, rigour and situated in the literature. With a few grammatical typos, it reads as a thread of different perspective, with little grounding in HCI and related field.

Lastly, in HCI, there is a movement towards ideas about participatory design, user-centred design, value-sensitive design and so on. A utilisation of these perspectives in framing the research ideas would have done more good to the paper than proposing a new design space for visualisation of user-generated data. From the guidelines outlined in section 9, it is hard to pinpoint new learning that the paper provides to the visualisation of subsequent design practices apart from restating well-known design insights. There is the question of how the data and the proposed guidelines might bring about some ‘implications for design’ (Dourish, 2006) and practice. Although the issues of implication for design has been misunderstood and widely misrepresented, what the proposed design guideline sought to point to might be regarded as some form of outlining implications for a design practice that is minimal and non-representative. This makes the paper weak, lacking impactful significance, and thus leaning would not argue strongly towards acceptance.   I would encourage the authors to situate the research questions into the broader literature and determine whether they fit into some of the well-established methods informing the designing of health-related technologies.

---

### Official Review · AnonReviewer1 · 2020-01-09
**A deep dive on visualization needs for patient & physician health data sharing**

**Confidence:** 4
**Rating:** 6

**Review:**

This work reports on a qualitative investigation of how visualization can support patients to share personal health data with health care providers. It entails a formative study with healthcare providers, interviews and prototype design of health data visualization for 8 patients, and follow-up interviews with physicians. The work concludes with a set of design guidelines that summarize findings from all three studies.

I found this a very interesting read. The detailed discussion of individual patient stories is very illustrative of the design challenges around this critically relevant domain. The study methodology is thorough, with iterations that consistently build on previous findings, and covers perspectives on both the patient and physician sides. This methodology could inspire similar studies in patient/doctor communication. The paper is generally well written, with a solid motivation and grounding in related work. It also provides a clear outline of the design guidelines; while most of them are derived from the formative study, they are fairly actionable, and contextually validated by the deep dive on patient stories.

The work also has a number of issues, although they can be largely addressed with further clarifications and better restructuring.

The most critical one is that while the paper is framed as covering BOTH patient and healthcare provider's perspectives, there is a disproportionate focus on the healthcare provider side. The designs were not discussed with patients, not their perspectives on the design contrasted with physician's feedback. Patients and physicians often have competing objectives, and this work stopped short of informing those. While I find there's sufficient contribution to the paper in its current form, the focus on the healthcare provider side should be much more clearly communicated from the Introduction onwards.

Second, I missed details on the overall design rationale behind the choices of visualization design beyond a general approach of "taking user's data collection & sharing needs into account". There were some cases that could arguably be supported by the same visualization type/style, and yet, design were really diverse and it wasn't clear to what extent they *had* to be. Was that a deliberate choice? How did you narrow down what particular use cases to address for each patient? How many designers were involved? What made you discard design alternatives? Were you explicitly seeking variety in the designs to enrich discussion with physicians? A biased focus on design diversity during the design process weakens the paper's criticism against "one-size-fits all" approaches, specially given the lack of validation with patients.

Finally, this work has fairly limited generalizability. While (a) this is less of a concern with qualitative studies and (b) recruitment in the medical space is famously challenging, a sample of 3 physicians is a bit too minimal to provide authoritative design guidance. This is evident enough with the lack of consensus on a number of topics in the interview discussions (e.g. whether or not allow for correlations to be drawn for patient views), and the limited impact of physician interviews on the design guidelines. I appreciate the statement that "these guidelines are not exhaustive", but again, I missed a more candid discussion of such challenges in a Limitations section.

In summary, this is an interesting read, albeit with some issues and limited generalizability of which many can be addressed with restructuring and clarifications. And while findings are not exactly new or ground breaking, the contextual richness provides a deeper level of understanding on these well known issues, and may serve as a good reference for those seeking concrete examples of patient cases.


***** Other Improvements

- *Significantly* improve resolution and usability of Figure 1. I would strongly recommend to bring each individual patient column closer to their respective visualization designs subsection, ideally on the same page.
- I missed more concrete details on the coding strategy. E.g., did you use open + selective coding, how many independent coders, etc.
- It seems likely but unclear that C1 and C2 from interviews were the same as two physicians in the as formative study. Clarify.
- It is unclear who the third "healthcare provider with 9y experience that supports patients monitoring data" is. A physician, a nurse, a nutritionist, health counsellor? Clarify role. Also important to note that these healthcare providers may have different objectives compared to physicians when it comes to leveraging patient data, which might be worth discussing.
- Fix typos: (-remove) (+add)
-- Page 2:
    "In (-the) this section..."
-- Page 3:
    "...we interviewed eight patients (+who?) actively collect health data..."
    "...we use (-the) grounded theory [46] to (-analysis)(+analyse) the data"
-- Page 7:
    "...in form of seven rings (days) (Fig. 1 - column P#6 - (-first) (+second) row)"

---

### Meta-Review · Area_Chair1 · 2020-01-12

**Recommendation:** Accept
**Confidence:** 4

**Metareview:**

Reviewers acknowledged that the paper is interesting, well-written, and relevant to the community. They also highlighted the thoroughness of the methodology employed.

However, they also highlighted some issues with the paper that the authors need to address: Below, I summarize the key issues. However, encourage the authors to read through the reviews carefully and address all issues highlighted by individual reviewers:

-	The paper claims to cover both patient and healthcare provider's perspectives, however, there is a disproportionate focus on the healthcare provider side [R1].
-	Lack of a clear rationale behind the choice of visualization presented in this paper [R1]
-	Limited generalizability of the findings [R1, R2]
-	Lack of clear articulation of the research problem and question in line with how they are situated within the literature [R2].
-	Failed to demonstrate how the authors showed some sensitivity towards participants with some form of a chronic condition [R2].
-	Some redundant and/or long-winded discussions [R3]
-	Some conflicting narratives and recommendations [R3]

Despite the shortcoming and highlighted weaknesses, the reviewers believe the paper hold some potential. I also believe that, although the issues highlighted are very important and must be addressed in the final version, they do not require significant changes in the paper. Hence, I recommend that the paper be accepted.

---

### Decision · Program_Chairs · 2020-01-12

Accept